# Reference intervals for common clinical chemistry parameters in healthy adults of Northeast Ethiopia

**Temesgen Fiseha**[1]*, **Ermiyas Alemayehu**[1], **Ousman Mohammed Adem**[1], **Bruktawit Eshetu**[1], **Angesom Gebreweld**[2]

**1** Department of Clinical Laboratory Science, College of Medicine and Health Sciences, Wollo University, Dessie, Ethiopia, **2** Department of Medical Laboratory Science, College of Health Sciences, Mekelle University, Mekelle, Ethiopia

* temafiseha@gmail.com

**Data Availability Statement:** All relevant data are within the paper and its Supporting Information files.

## Abstract

### Background

Clinical chemistry reference intervals are important tools for health evaluation, diagnosis, prognosis and monitoring adverse events. Currently used reference intervals in most African countries including Ethiopia are mainly derived from Western populations, despite studies reporting differences that could lead to incorrect clinical decisions. The aim of this study was to establish reference intervals for commonly used clinical chemistry parameters for healthy adults in Northeast Ethiopia.

### Methods

A community based cross-sectional study was conducted among 328 apparently healthy adults between the ages of 18 and 57 years. Blood samples were collected for clinical chemistry analysis using Dirui CS-T240 auto-analyzer and serological testing to screen the population. Medians and 95% reference intervals were computed using non-parametric method according to the Clinical and Laboratory Standards Institute guideline. The Mann–Whitney U test was used to compare reference values between males and females.

### Results

Reference intervals established were: ALT 11.2–48.0 U/L, AST 16–60 U/L, ALP 53–342.3 U/L, total protein 5.4–8.9 mg/dL, total bilirubin 0.1–1.23 mg/dL, glucose 65–125 mg/dL, total cholesterol 69–213 mg/dL, triglycerides 46–207 mg/dL, creatinine 0.3–1.2 mg/dL and urea 9.5–46.3 mg/dL. Significant sex-differences were observed for ALT, AST, ALP, total cholesterol, triglycerides, creatinine and urea. We found that the established reference intervals substantially differ from the reference ranges currently in use. Up to 43.1% of apparently healthy adults are considered as having abnormal test values on the bases of the currently in use reference ranges. If the reference values from the United States based intervals were applied to the study population, 81.8% would have been classified as having abnormal laboratory test results.

**Funding:** This study was financial supported by Wollo University, www.wu.edu.et. The recipient is Mr. Temesgen Fiseha. The role of the funding institute is close follow up of the research activity through monitoring and evaluation.

**Competing interests:** The authors have declared that no competing interests exist

**Abbreviations:** ALP, alkaline phosphatase; ALT, alanine amino transferase; AST, aspartate amino transferase; CRP, C-reactive protein; HBsAg, Hepatitis B surface antigen; HBV, hepatitis B virus; HCV, hepatitis C virus; HIV, human immune deficiency virus; mg/dl, milligram per deciliter; NA, not available; QC, quality control; U/L, units per liter.

## Conclusions

Local population-specific reference intervals were established for commonly used clinical chemistry parameters in adult population of Northeast Ethiopia. Although further study is needed, these reference intervals may have the potential to facility the decision-making process based on laboratory test results in this population.

## Introduction

Reference intervals are important tools for accurate interpretation of results from laboratory testing [1, 2]. They can be used in clinical care and clinical trials for appropriate assessment of health status, monitoring disease progression, and reporting of possible toxicity and adverse events. Reference intervals relevant to the population of interest, for which a particular test will be applied, are required to interpret values of laboratory test results [3–5]. However, most laboratories in sub-Saharan Africa refer to reference values provided by the package inserts of *vitro* diagnostic manufacturers or external literature sources instead of determining population-specific intervals [6, 7]. Studies have documented that applying these reference intervals would result in inaccurate interpretation of laboratory results leading to misclassification and misdiagnosis of healthy African people [8–11]. This strongly advocates the need for the establishment and use of local population-specific reference intervals for common clinical laboratory test parameters.

Clinical chemistry reference intervals are most important elements for health evaluation, diagnosis, staging, prognostication and monitoring adverse events. Currently used reference intervals for these parameters in most African countries are mainly derived from Western populations [12]. Differences in reference values of chemistry parameters have been reported compared to values of healthy Africans [8, 9, 11]. Moreover, variations in reference intervals have been reported across different populations of Africa [11, 13–15]. Factors such as age, gender, body fat distribution pattern, ethnic origin, geographic locations, diet and analytical methods can affect the distribution of biochemical analytes [16, 17]. It is therefore important that all laboratories in African countries should determine and maintain their own reference intervals for such parameters. This is even more important given the number of clinical trials and persons receiving clinical services is expected to increase substantially in sub-Saharan Africa [6, 12].

In Ethiopia, there is limited data regarding locally established reliable reference intervals and clinical chemistry reference intervals used currently in clinical laboratories are values obtained from Western populations [18]. The few studies that have been undertaken showed differences in clinical chemistry reference values compared to Western based values [15, 19, 20]. These differences have the potential to create confusion in the interpretation of test results and impacts clinical decision-making. It is critical to establish population-specific clinical chemistry reference intervals in Ethiopia for the accurate assessment of patient health and to ensure the safe conduct of medical researches. The aim of this study was to establish reference intervals for commonly used clinical chemistry parameters specific to adult populations of Northeast Ethiopia.

## Materials and methods

### Study design and population

A community based cross-sectional study was conducted from April 2019 to January 2021 among apparently healthy adults living in South Wollo Zone, Northeast Ethiopia. The study

population comprised of adult male and females from Dessie town and surrounding areas of Tita, Gerado and Borumeda. Dessie is the capital city of South Wollo Zone in the Amhara Regional state, located 401 km Northeast of the capital Addis Ababa, Ethiopia. Tita, Gerado and Borumeda are rural communities of Dessie located within Dessie specialized referral hospital catchment areas in South Wollo Zone, Northeast Ethiopia.

## Sample size and sampling techniques

The sample size of this study was determined according to the Clinical and Laboratory Standards Institute (CLSI) guideline, which recommends a minimum of 120 healthy individuals for a 95th percentile reference interval determination with 95% confidence intervals for each partition group (e.g. sex) [21]. However, according to previous studies in other African countries [22], about 30% of apparently healthy population do not qualify for reference interval determination for various reasons when tested for the common viral infections and syphilis. Considering this, to reach the CLSI recommended total sample size of 240 for the reference interval determination, a total of 344 healthy volunteers were enrolled. The study population was selected using the priori convenient sampling technique from the community residing in Dessie town and surrounding areas of South Wollo Zone, Northeast Ethiopia. Considering altitude and residence difference; Dessie town, and Tita, Gerado and Borumeda rural communities were taken purposively from the study area. The determined sample size was distributed proportional to their population size of each selected community. From the selected communities, sub-communities were determined conveniently on the bases of easy to reach and suitability for blood sample transportation to the hospital laboratory, where biochemical analysis and serological testing was conducted. Finally, individuals who met the study eligibility criteria were included in the study from each community until the required number was achieved.

## Eligibility criteria

Participants were identified and included based on stringent inclusion and exclusion criteria. Each participant received a review of medical history, a complete physical examination, and testing for C-reactive protein (CRP), human immune deficiency virus (HIV), hepatitis, syphilis and pregnancy (for females). Apparently healthy adults aged 18 years and above who have lived in the areas for at least 5 years were finally included in the study. Individuals with any of the following were excluded from the reference group: known chronic diseases (diabetes mellitus, hypertension, chronic renal insufficiency, ischemic heart disease, anemia, thyroid disorders, and liver diseases); history of blood donation in the last 6 months; blood transfusion in the previous year; intake of pharmacologically active agents and all prescription drugs; and a positive result from the screening tests (CRP, HIV, hepatitis B surface antigen (HBsAg), anti-hepatitis C virus (HCV) antibody, syphilis and Human Chorionic Gonadotropin [HCG] for females).

## Data collection and laboratory analysis

Study participants were invited to come to the nearby health institution, where individual consenting, screening and blood collections were carried out. Individuals who agreed to give written consent after being informed about the purpose of the study and associated risks underwent a detailed medical history and physical examinations by trained nurses. A structured, pre-tested questionnaire was used to collect socio-demographic and medical histories. Physical examination was performed using calibrated equipment's and standardized techniques on site.

Five ml of venous blood samples were collected using plane tubes between 8 and 11 am in the morning after an overnight fast. Fasting blood glucose and urine HCG testing were performed on site. The collected blood samples were allowed to clot for 60 minute at room temperature, then centrifuged for 5 minute at 2500 rpm, and serum separated. The serum samples were transported in sealed boxes to Dessie specialized referral hospital laboratory and were processed within 4 hours of collection. Biochemical analysis was done using Dirui CS T240 auto-analyzer (Dirui Industrial Company, China). The following commonly used clinical chemistry parameters were measured according to the manufacturer's instructions: alanine aminotransferase (ALT, IFCC method, without pyridoxal-5′ phosphate), aspartate amino-transferase (AST, IFCC method, without pyridoxal-5′-phosphate), alkaline phosphatase (ALP, IFCC method), total protein (biuret method), total bilirubin (surfactant/diazo salt method), glucose (glucose oxidase method), total cholesterol (enzymatic method), triglycerides (oxidase method), creatinine (enzymatic method) and urea (enzymatic method).

## Quality assurance

All processes of pre-analysis, analysis and post-analysis were conducted in accordance with Good Clinical Laboratory Practices (GCLP) using standard operating procedures (SOPs). Analysis of samples was done after proper standardization of the instrument with the help of calibrators and internal controls. Internal quality control (QC) was performed using two levels of QC sera (normal and abnormal) for each analyte under the study. Daily internal QC was performed with each subsequent batch of analyses. The hospital laboratory also takes part in the External Quality Assessment (EQA) program run by the Ethiopian Public Health Association (EPHI).

## Statistical analysis

Data were checked for completeness, cleared and entered into EpiData version 3.1 software (Epidata Association, Odense, Denmark). Data analyses were performed by using SPSS version 25 software (SPSS Inc., Chicago, IL, USA) and MedCalc version 20.027 software (Ostend, Belgium). The normal distribution was tested by the Kolmogorov-Smirnov test. The Dixon method was used to detect and eliminate extreme values as outliers [23]. The mean, median and nonparametric 95% reference intervals (2.5th and 97.5th percentiles) were calculated for each clinical chemistry parameter. The 90% confidence intervals (CI) for the lower and upper limits of the reference intervals were also determined in accordance with the CLSI guidelines [21]. The Mann–Whitney U test was used to test for differences in reference values by gender. A P-value of < 0.05 was considered statistically significant.

## Ethical considerations

Ethical approval was obtained from the Institutional Review Board of College of Medicine and Health Sciences, Wollo University. Written informed consent was obtained from each study participants. All participants diagnosed for any illness were treated accordingly. Information obtained at any course of the study was kept confidential.

## Results

Of 344 participants consented for the study through a priori convenient sampling technique, 16 were excluded based on serological tests. A total of 328 (164 males and 164 females) apparently healthy adults were included in the final analysis for establishing reference intervals of commonly used clinical chemistry parameters for adult populations of South Wollo Zone, Northeast

Ethiopia. The mean age of the study participants was 29.2 ± 8.2 years (male = 29.8 ± 8.1 years and female = 28.6 ± 8.2 years), ranging from 18 to 57 years. Majority of (59.4%) the participants were from urban communities and the mean body mass index was 23.2 ± 3.1 Kg/m$^2$.

The mean, median and 95% reference intervals with 90% CI of the lower and upper limits for the clinical chemistry parameters are shown in Table 1. The median and 95% reference intervals for males and females, respectively were: ALT 25 (14–48 U/L) and 21 (8–49 U/L), AST 38 (19–63 U/L) and 30 (16–59 U/L), ALP 204 (55–343 U/L) and 182.5 (50–326 U/L), total protein 6.8 (5.5–8.1 g/dL) and 7.1 (5.2–9.0 g/dL), total bilirubin 0.45 (0.1–1.3 mg/dL) and 0.40 (0.1–1.2 mg/dL), glucose 97 (65–126 mg/dL) and 95 (60–119 mg/dL), total cholesterol 160 (97–208 mg/dL) and 171 (69–216 mg/dL), triglycerides 146 (44–220 mg/dL) and 133 (46–207 mg/dL), creatinine 0.8 (0.4–1.3 mg/dL) and 0.7 (0.3–1.1 mg/dL), and urea 28 (13–47 mg/dL) and 24 (8–42 mg/dL).

**Table 1.** **The mean, median and 95% reference intervals of clinical chemistry parameters for healthy adults in Northeast Ethiopia.**

| Parameters | Sex | Mean | Median | 95% RI | 95% RI | | P-value |
|---|---|---|---|---|---|---|---|
| | | | | | 90% CI (lower limit) | 90% CI (upper limit) | |
| ALT (U/L) | Combined | 25.2 | 23.0 | 11.2–48.0 | 7.0, 14.0 | 43.0, 55.0 | 0.002 |
| | Males | 26.7 | 25.0 | 14.0–48.9 | 6.0, 15.0 | 43.0, 60.0 | |
| | Females | 23.8 | 21.0 | 7.9–48.0 | 6.0, 13.0 | 40.0, 53.0 | |
| AST (U/L) | Combined | 36.8 | 33.0 | 16.0–60.0 | 15.0, 21.0 | 60.0, 63.0 | < 0.001 |
| | Males | 39.6 | 38.0 | 18.9–63.0 | 16.0, 21.0 | 60.0, 69.0 | |
| | Females | 34.0 | 30.0 | 15.9–59.0 | 15.0, 16.0 | 54.0, 59.0 | |
| ALP (U/L) | Combined | 196.2 | 195.0 | 53.0–342.3 | 49.0, 55.0 | 320.0, 346.0 | 0.009 |
| | Males | 208.6 | 204.0 | 55.0–343.4 | 48.0, 105.0 | 330.0, 359.0 | |
| | Females | 183.9 | 182.5 | 50.0–326.3 | 45.0, 54.0 | 309.0, 345.0 | |
| Total protein (g/dl) | Combined | 6.9 | 7.0 | 5.4–8.9 | 5.2, 5.5 | 8.9, 9.0 | 0.453 |
| | Males | 6.9 | 6.8 | 5.5–8.1 | 5.2, 5.6 | 7.8, 8.7 | |
| | Females | 6.9 | 7.1 | 5.2–9.0 | 4.8, 5.4 | 8.9, 9.0 | |
| Total bilirubin (mg/dl) | Combined | 0.53 | 0.40 | 0.10–1.23 | 0.10, 0.11 | 1.20, 1.30 | 0.500 |
| | Males | 0.53 | 0.45 | 0.10–1.30 | 0.10, 0.11 | 1.20, 1.30 | |
| | Females | 0.53 | 0.40 | 0.10–1.20 | 0.10, 0.12 | 1.18, 1.22 | |
| Glucose (mg/dl) | Combined | 94 | 94 | 60–125 | 60, 64 | 119, 130 | 0.999 |
| | Males | 96 | 97 | 65–126 | 61, 69 | 124,130 | |
| | Females | 94 | 95 | 60–119 | 59, 61 | 116, 122 | |
| Total cholesterol (mg/dl) | Combined | 159 | 160 | 69–213 | 69, 80 | 210, 217 | 0.042 |
| | Males | 157 | 160 | 97–208 | 76, 100 | 204, 217 | |
| | Females | 161 | 171 | 69–216 | 61, 79 | 213, 227 | |
| Triglycerides (mg/dl) | Combined | 136 | 140 | 46–207 | 36, 61 | 203, 232 | 0.029 |
| | Males | 140 | 146 | 44–220 | 12, 60 | 203, 238 | |
| | Females | 132 | 133 | 46–207 | 36, 61 | 203, 236 | |
| Creatinine (mg/dl) | Combined | 0.76 | 0.80 | 0.30–1.20 | 0.30, 0.40 | 1.20, 1.30 | 0.002 |
| | Males | 0.81 | 0.80 | 0.40–1.30 | 0.40, 0.50 | 1.20, 1.30 | |
| | Females | 0.71 | 0.70 | 0.30–1.11 | 0.30, 0.40 | 1.10, 1.20 | |
| Urea (mg/dl) | Combined | 26.3 | 25.5 | 9.5–46.3 | 6.0, 14.0 | 45.0, 48.0 | < 0.001 |
| | Males | 28.4 | 28.0 | 13.0–47.1 | 6.0, 15.0 | 45.0, 48.0 | |
| | Females | 24.3 | 24.0 | 8.0–41.5 | 6.0, 10.0 | 36.1, 46.1 | |

P-value: Mann–Whitney U test for males versus females; ALT: alanine aminotransferase; ALP: alkaline phosphatase; AST: aspartate aminotransferase; CI: confidence interval; mg/dl: milligram per deciliter; RI: reference interval; U/L: units per liter

The median values of ALT, AST, ALP, triglycerides, creatinine and urea in males were significantly higher than in females, while females had significantly higher median total cholesterol values compared to males ($P < 0.05$). The upper limits of the reference intervals in males were higher ($> 10\%$) than in females for creatinine and urea, and lower ($> 10\%$) than in females for total protein. The lower limits of the reference intervals in males were higher ($> 10\%$) than females for ALT, AST, total cholesterol, creatinine and urea (Table 1).

Tables 2 and 3 shows the comparison of the clinical chemistry reference intervals obtained in this study with the reference intervals currently in use and those of other studies in Africa and the United States (US). The reference intervals obtained for some parameters in this study were comparable with other studies in the region. However, there are marked differences ($> 10\%$) in the lower and/or upper limits of the reference intervals obtained for ALT, AST, ALP, glucose, triglycerides, total cholesterol and creatinine compared to the reference ranges currently in use. The upper reference limits for ALT, AST, ALP, glucose and triglycerides

**Table 2. Comparison of clinical chemistry reference intervals obtained in this study with previous studies in Ethiopia and the values currently in use.**

| Parameters | Sex | Present study | Northwest Ethiopia (Gojjam) [19] | Amhara Regional State [15] | Southwest Ethiopia [20] | Currently in use |
|---|---|---|---|---|---|---|
| ALT (U/L) | C | 11–48 | 6.0–43.0 | 5.0–39.0 | 11.0–54.0 | 0–42 |
|  | M | 14–49 | 6.0–44.6 | 5.1–42.9 | 11.2–56.0 | |
|  | F | 7.9–48 | 3.0–30.0 | 4.3–37.0 | 10.1–54.0 | |
| AST (U/L) | C | 16–60 | 9.0–38.0 | 11.0–46.0 | 12.0–59.0 | 0–37 |
|  | M | 19–63 | 10.5–39.0 | 12.1–46.9 | 13.0–59.5 | |
|  | F | 16–59 | 6.0–32.1 | 10.0–43.8 | 12.0–59.9 | |
| ALP (U/L) | C | 53–342 | 52.4–237.0 | 87.0–451.3 | 63.0–376.0 | 0–270 |
|  | M | 55–343 | 55.3–237.2 | 77.2–475.8 | 55.8–362.9 | |
|  | F | 50–326 | 49.0–236.0 | 89.0–381.0 | 70.4–384.4 | |
| Total protein (g/dl) | C | 5.4–8.9 | 5.3–8.6 | 5.7–9.6 | 4.4–11.6 | 5.3–8.7 |
|  | M | 5.5–8.1 | 5.3–8.67 | 5.7–9.7 | 4.0–11.4 | |
|  | F | 5.2–9.0 | 5.32–8.60 | 5.6–9.5 | 4.6–11.7 | |
| Total bilirubin (mg/dl) | C | 0.1–1.23 | 0.26–2.18 | 0.1–1.1 | NA | 0–1.2 |
|  | M | 0.1–1.3 | 0.27–2.18 | 0.1–1.2 | NA | |
|  | F | 0.1–1.2 | 0.21–2.18 | 0.08–0.9 | NA | |
| Glucose (mg/dl) | C | 60–125 | NA | NA | NA | 70–110 |
|  | M | 65–126 | NA | NA | NA | |
|  | F | 60–119 | NA | NA | NA | |
| Total cholesterol (mg/dl) | C | 69–213 | NA | 80.4–206.6 | 55.0–276.0 | 0–200 |
|  | M | 97–208 | NA | 78.1–211.8 | 52.1–252.2 | |
|  | F | 69–216 | NA | 83.6–202.7 | 58.0–286.4 | |
| Triglycerides (mg/dl) | C | 46–207 | NA | 36.0–215.6 | 41.0–264.0 | 0–165 |
|  | M | 44–220 | NA | 36.0–221.9 | 41.3–275.8 | |
|  | F | 46–207 | NA | 35.3–201.5 | 41.0–261.2 | |
| Creatinine (mg/dl) | C | 0.3–1.2 | 0.23–1.22 | 0.47–1.12 | 0.32–1.32 | 0.5–1.2 |
|  | M | 0.4–1.3 | 0.2–1.29 | 0.48–1.13 | 0.3–1.4 | |
|  | F | 0.3–1.11 | 0.25–1.08 | 0.47–1.09 | 0.3–1.3 | |
| Urea (mg/dl) | C | 9.5–46.3 | NA | 11.0–41.0 | 4.6–35.0 | 10–50 |
|  | M | 13.0–47.1 | NA | 12.0–43.0 | 4.6–34.5 | |
|  | F | 8.0–41.5 | NA | 10.0–38.7 | 4.5–35.8 | |

ALT: alanine aminotransferase; ALP: alkaline phosphatase; AST: aspartate aminotransferase; C: combined for both males and females; F: females; M: males; mg/dl: milligram per deciliter; NA: not available; U/L: units per liter

**Table 3. Comparison of clinical chemistry reference intervals obtained in this study with other African studies and these of the United States.**

| Parameters | Sex | Present study | Kenya [11] | Tanzania [14] | Nigeria [27] | Ghana [13] | Uganda [10] | Textbook [25, 26] | USA [24] |
|---|---|---|---|---|---|---|---|---|---|
| ALT (U/L) | C | 11–48 | 9.6–52 | 8–48 | NA | 7–51 | 6.6–42.8 | | 0–35 |
| | M | 14–49 | 10.8–53.9 | 9–55 | 17.3–48.4 | 8–54 | 7.2–43.3 | 0–45 | NA |
| | F | 7.9–48 | 8.6–47 | 7–45 | 19–38 | 6–51 | 5.3–39.9 | 0–34 | NA |
| AST (U/L) | C | 16–60 | 13.8–42.3 | 14–48 | NA | 14–51 | 12.3–34.8 | | 0–35 |
| | M | 19–63 | 14.9–45.3 | 15–53 | 26–49.4 | 17–60 | 13.2–35.9 | 0–35 | NA |
| | F | 16–59 | 13.1–38.1 | 14–35 | 22–58.4 | 13–48 | 11.4–28.8 | 0–31 | NA |
| ALP (U/L) | C | 53–342 | NA | 45–158 | NA | 85–241 | 44–151 | | 30–120 |
| | M | 55–343 | NA | 45–170 | NA | 101–355 | 42–159 | 53–128 | NA |
| | F | 50–326 | NA | 45–155 | NA | 82–293 | 47–160 | 42–98 | NA |
| Total protein (g/dl) | C | 5.4–8.9 | NA | 6.6–8.7 | NA | 5.1–8.7 | 6.6–8.9 | 6.3–8.3 | 5.5–8.0 |
| | M | 5.5–8.1 | NA | 6.7–8.5 | NA | 4.7–8.6 | 6.5–8.9 | | NA |
| | F | 5.2–9.0 | NA | 6.6–8.6 | NA | 5.5–8.7 | 6.8–9.0 | | NA |
| Total bilirubin (mg/dl) | C | 0.1–1.23 | 0.29–2.33 | 0.3–2.48 | NA | 0.17–1.51 | 0.4–2.5 | 0.2–1.0 | 0.3–1.0 |
| | M | 0.1–1.3 | 0.33–2.51 | 0.35–2.44 | 0.2–1.0 | 0.22–1.87 | 0.4–2.6 | | NA |
| | F | 0.1–1.2 | 0.26–1.57 | 0.26–1.83 | 0.02–0.62 | 0.16–1.56 | 0.3–1.9 | | NA |
| Glucose (mg/dl) | C | 60–125 | NA | 52.6–94.1 | NA | 64.8–115.2 | NA | 74–106 | 75.6–115.2 |
| | M | 65–126 | NA | 51.8–95.4 | 66.6–142.2 | 63.0–113.4 | NA | | NA |
| | F | 60–119 | NA | 54.0–91.1 | 79.2–172.9 | 66.6–118.0 | NA | | NA |
| Total cholesterol (mg/dl) | C | 69–213 | NA | 95.5–213.8 | NA | 77.3–208.1 | 91–233 | 140–200 | <200 |
| | M | 97–208 | NA | 89.7–219.3 | 123.7–205 | 69.6–193.3 | 90–235 | | NA |
| | F | 69–216 | NA | 109–212.7 | 120–216.6 | 81.2–216.6 | 100–230 | | NA |
| Triglycerides (mg/dl) | C | 46–207 | NA | 34.5–255.1 | NA | 35.4–194.9 | 39–281 | 65–157 | <160 |
| | M | 44–220 | NA | 34.5–266.6 | 62–194.9 | 35.4–194.9 | 39–299 | | NA |
| | F | 46–207 | NA | 33.7–193.1 | 53.1–186 | 35.4–186 | 34–206 | | NA |
| Creatinine (mg/dl) | C | 0.3–1.2 | NA | 0.47–1.02 | NA | 0.55–1.33 | 0.5–1.2 | | <1.5 |
| | M | 0.4–1.3 | 0.7–1.2 | 0.54–1.09 | 0.86–1.26 | 0.63–1.35 | 0.6–1.2 | 0.9–1.3 | NA |
| | F | 0.3–1.11 | 0.58–1.03 | 0.45–0.92 | 0.71–1.33 | 0.53–1.24 | 0.5–0.9 | 0.6–1.1 | NA |
| Urea (mg/dl) | C | 9.5–46.3 | NA | 9.1–29.5 | NA | 5.4–34.2 | 9.9–33.2 | 12.6–42.6 | 21.4–42.9 |
| | M | 13.0–47.1 | NA | 9.3–29.8 | 13.2–28.8 | 5.4–37.2 | 10.1–33.9 | | NA |
| | F | 8.0–41.5 | NA | 8.8–27.4 | 15.0–34.8 | 5.4–32.4 | 9.4–30.2 | | NA |

ALT: alanine aminotransferase; ALP: alkaline phosphatase; AST: aspartate aminotransferase; C: combined for both males and females; F: females; M: males; mg/dl: milligram per deciliter; NA: not available; U/L: units per liter

obtained in this study were significant higher (> 10%) than the reference values currently being used. When compared with the US based values mostly used as the standard reference interval comparison for most studies [24–26], the upper limits of the reference intervals determined for our population were higher except for creatinine.

The proportion of adults in our area who would have been considered as having abnormal test results when compared with the reference values currently in use and those of the US [24] are presented in Table 4. Up to 43.1% of apparently healthy adults are considered as having abnormal test values above the upper limits of the currently in use reference intervals, with a higher proportion of out of range values observed for AST. If the upper limit from the US reference intervals were applied to the entire study population, 81.8% would have had abnormal ALP. These proportions are 43.4%, 31.1%, 19.2%, 16.4%, 13.8%, 7.5% and 4.4% for AST, triglycerides, total cholesterol, ALT, total bilirubin, glucose and urea, respectively.

**Table 4. Clinical chemistry out of range values based on comparison with the reference intervals currently in use and those of the United States.**

| Parameters | Present study | Currently in use | | | Out of range comparison USA | | |
|---|---|---|---|---|---|---|---|
| | | n | % | 95% reference interval | n | % | 95% reference interval |
| ALT (U/L) | 11–48 | 21 | 6.6 | 0–42 | 52 | 16.4 | 0–35 |
| AST (U/L) | 16–60 | 137 | 43.1 | 0–37 | 138 | 43.4 | 0–35 |
| ALP (U/L) | 53–342 | 41 | 12.9 | 0–270 | 260 | 81.8 | 30–120 |
| Total protein (g/dl) | 5.4–8.9 | 00 | 0.0 | 5.3–8.7 | 14 | 4.4 | 5.5–8.0 |
| Total bilirubin (mg/dl) | 0.1–1.23 | 34 | 10.7 | 0–1.2 | 44 | 13.8 | 0.3–1.0 |
| Glucose (mg/dl) | 60–125 | 60 | 18.9 | 70–110 | 24 | 7.5 | 75.6–115.2 |
| Total cholesterol (mg/dl) | 69–213 | 61 | 19.2 | 0–200 | 61 | 19.2 | <200 |
| Triglycerides (mg/dl) | 46–207 | 91 | 29.6 | 0–165 | 99 | 31.1 | <160 |
| Creatinine (mg/dl) | 0.3–1.2 | 14 | 4.4 | 0.5–1.2 | 00 | 0.0 | <1.5 |
| Urea (mg/dl) | 9.5–46.3 | 00 | 0.0 | 10–50 | 14 | 4.4 | 21.4–42.9 |

## Discussion

The clinical utility of clinical chemistry laboratory tests deepens crucially on the availability of accurate reference intervals for their interpretation. Due to lack of locally established reference intervals, most clinical laboratories in Ethiopia currently rely on reference values mainly derived from Western populations [18]. In light of these critical gaps, this study established reference intervals for commonly used clinical chemistry parameters for healthy adult populations of Northeast Ethiopia.

The reference intervals of ALT and AST in this study are comparable with reports from Southwest Ethiopia [20] and Tanzania [14]. However, the upper reference limits of ALT and AST in our study are lower than those reported from Kenya [11] and Ghana [13], and higher than those from Northwest Ethiopia [19], Amhara Regional State [15], Uganda [10], Textbooks [25, 26] and USA [24]. The reference values of serum ALT and AST vary in different studies mostly related to characteristics of the reference populations including age, sex, body fat and its distribution, race/ethnicity, dietary pattern and serum lipid levels [16, 28]. Normal serum liver enzyme levels can also vary with variations in lifestyle, environmental factors and analytical methods [25, 26, 29]. The higher median ALT and AST values in males compared to females in this study are consistent with reports from previous studies in Ethiopia [15, 19, 20], Uganda [10], Ghana [13] and Tanzania [14]. The sex-specific differences in serum liver enzymes may be partly related to the differences in muscle mass, body fat and fat-muscle distribution due to the effects of sex hormones [16, 30].

The ALP reference intervals in this study are lower than reports from Amhara Regional State [15] and Southwest Ethiopia [20], but higher than those from Northwest Ethiopia [19], Uganda [10], Ghana [13], Tanzania [14], Textbooks [25, 26] and USA [24]. The difference in the reference values of ALP may be caused by differences in the age, sex, ethnic, hormonal status, eating habits and blood type distribution of the reference population, or due to geographical and analytical differences [28, 31]. Significant differences in serum ALP values were observed between males and females, with males having higher median values. The sex-related differences observed in serum ALP values are consistent with reports from other parts of Ethiopia [15, 19, 20], Uganda [10], Ghana [13] and Tanzania [14], and calls for separate reference intervals to be used for males and females. Sex-differences in serum ALP could be attributed to protein eating habit, body fat distribution, and to the differences in sex hormones [31, 32].

The lower limit of the reference interval for total protein in this study was comparable with reports from Northwest Ethiopia [19] and USA [24]. But, it was lower than those from Amhara

Regional State [15], Uganda [10], Tanzania [14] and Textbooks [25, 26], and higher than those from Southwest Ethiopia [20] and Ghana [13]. The absence of sex-differences in total protein value is consistent with the majority of reports in African adults [13, 14, 19]. The upper reference limit for total bilirubin in this study was lower than those from Northwest Ethiopia [19], Uganda [10], Kenya [11], Ghana [13] and Tanzania [14], and higher than those from Amhara Regional State [15], Textbooks [25, 26] and USA [24]. Reports on sex-differences in total bilirubin values are mixed: some reports show no differences between sex groups, as we did [19], while others report higher values in males than in females [10, 11, 13–15, 27]. The reference interval for glucose in this study was comparable with reports from Ghana [13]; but higher than those from Tanzania [14], Textbooks [25, 26] and USA [24]. We noted no sex-differences in glucose values; consistent with results of previous studies [13, 14], but differs from reports from Nigeria [27]. The observed differences in reference intervals between studies could be related to variations in the age, sex, race, nutritional status and environmental factors [17, 28].

The reference intervals obtained for total cholesterol and triglycerides in this study are comparable with those reported from Amhara Regional State [15], Ghana [13] and Tanzania [14]; but lower than those from Southwest Ethiopia [20] and Uganda [10], and higher than those from Textbooks [25, 26] and USA [24]. These differences might be explained by the difference in the demographic, ethnicity, body fat distribution pattern and hormonal status of the population. Normal reference values of serum lipids may also be affected by dietary habits, lifestyle and environmental factors, or analytical differences [17, 33]. The median values for total cholesterol were found to be higher in females than in males, similar to what has been previously reported from Amhara Regional State [15], Southwest Ethiopia [20], Ghana [13] and Burkina Faso [34]. In this study, we found that serum triglycerides values in males were significantly higher than those in females; consistent with reports from other studies in Ethiopia [15, 20], Uganda [10] and Nigeria [27]. Sex-differences in serum lipids could be attributed to differences in body fat distribution between sexes due to effects of sex hormone patterns and sex-related genetic factors [35, 36].

The reference intervals for renal function parameters (creatinine and urea) in this study were comparable with those reported from Amhara Regional State [15], Northwest Ethiopia [19] and Uganda [10]. The intervals however were higher than those from Southwest Ethiopia [20], Ghana [13], Tanzania [14] and Kenya [37], and lower than those from Textbooks [25, 26] and USA [24]. The differences in reference intervals of serum creatinine and urea among studies might be due to demographic differences such as age and sex, muscle mass, race/ethnicity, dietary and lifestyle (physical exercise) habits, environmental and genetic variation [25, 38]. The sex-related differences in serum creatinine values observed in this study are in line with previous studies [10, 11, 13–15, 19, 20, 27, 37, 39] and this is due to the fact that because creatinine is a product of muscle catabolism, serum levels are higher in males [40]. The significantly higher values of urea in males compared to females in this study is supported by previous studies in Ethiopia [15, 20] and other African countries [10, 13, 27, 37]. Differences in protein intake habit and lean body mass between males and females might, in part, account for the sex-specific variations in urea levels [41].

Our results showed that there are apparent differences in the lower and/or upper limits of the reference intervals obtained for majority of the commonly used serum chemistry tests, compared to the reference values currently in use. The upper reference limits obtained for ALT, AST, ALP, glucose and triglycerides were higher (> 10%) than the laboratory normal ranges currently in use. On the basis of the currently in use reference values, up to 43.1% of apparently healthy adults in our area are considered as having abnormal results. This highlights the influence of local reference populations on normal serum chemistry reference values and support the need for the establishment of population-specific reference intervals as a lack

or inappropriate use of reference intervals may lead to adverse consequences including misdiagnosis, patient risk and higher health care costs, all of which impact the quality of patient healthcare [3, 42]. This is the reason why international regulatory bodies recommend every clinical laboratory establish their own reference intervals specific to the local population being served [4, 5, 21].

Most of the reference intervals obtained from this study were quite different from the US-based intervals [24], mostly used for screening and enrolment of participants in medical studies. If the US-based reference values were applied as standards for the interpretation of the normalcy of laboratory results, up to 81.8% of the study population would have been classified as having abnormalities. These findings are consistent with previous reports in the region [8–12, 14, 43], suggesting that Western derived reference values used in clinical studies may not be applicable to African populations. Using the US-based reference values, the proportion of otherwise clinically healthy subjects whose values were abnormal was up to 80% in Kenya [11] and up to 81% in Tanzania [14]. Thus, use of reference intervals representative of the population being investigated may help prevent misdiagnosis and unnecessary exclusions during screening [44, 45]. This is a major indication for the establishment of local population-specific reference intervals for accurate interpretation of laboratory test results to aid in the screening process and safety evaluation in medical studies.

In conclusion, the present study established reference intervals for commonly requested clinical chemistry parameters in apparently healthy Ethiopian adults. Significant sex-differences were observed for most parameters, except for total protein, total bilirubin and glucose. Our data support previous studies that Western derived chemistry reference values used in routine clinical care and during medical studies may not be applicable to African populations. Although further research is needed, the reference intervals established may have the potential to facility the decision-making process of a measured laboratory test results in this population.

## Supporting information

**S1 Dataset. The excel database used for this manuscript.**
(XLSX)

## Acknowledgments

The authors acknowledge the community members who took part in this study and the health staff at the Dessie Specialized Referral Hospital for their assistance in gathering the data.

## Author Contributions

**Conceptualization:** Temesgen Fiseha, Angesom Gebreweld.

**Data curation:** Temesgen Fiseha, Ermiyas Alemayehu, Ousman Mohammed Adem, Bruktawit Eshetu, Angesom Gebreweld.

**Formal analysis:** Temesgen Fiseha.

**Funding acquisition:** Temesgen Fiseha.

**Investigation:** Temesgen Fiseha, Ermiyas Alemayehu, Ousman Mohammed Adem, Bruktawit Eshetu.

**Methodology:** Temesgen Fiseha, Ermiyas Alemayehu, Bruktawit Eshetu, Angesom Gebreweld.

**Software:** Temesgen Fiseha, Angesom Gebreweld.

**Supervision:** Ermiyas Alemayehu, Ousman Mohammed Adem.

**Visualization:** Ousman Mohammed Adem.

**Writing – original draft:** Temesgen Fiseha.

**Writing – review & editing:** Temesgen Fiseha, Ermiyas Alemayehu, Ousman Mohammed Adem, Bruktawit Eshetu, Angesom Gebreweld.

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
