## [Decision Letter · Decision Letter 0]

21 Aug 2022

PONE-D-22-12892Reference intervals for common clinical chemistry parameters in healthy adults of Northeast EthiopiaPLOS ONE

Dear Dr. Fiseha,

Thank you for submitting your manuscript to PLOS ONE. After careful consideration, we feel that it has merit but does not fully meet PLOS ONE’s publication criteria as it currently stands. Therefore, we invite you to submit a revised version of the manuscript that addresses the points raised during the review process.

We look forward to receiving your revised manuscript.

Kind regards,

Donovan Anthony McGrowder, PhD., MA., MSc

Academic Editor

PLOS ONE

Journal Requirements:

Additional Editor Comments:

Dear Dr. Fiseha,

Your manuscript “Reference intervals for common clinical chemistry parameters in healthy adults of Northeast Ethiopia” has been assessed by our reviewers. They have raised a number of points which we believe would improve the manuscript and may allow a revised version to be published in PLOS ONE. Their reports, together with any other comments, are below.

If you are able to fully address these points, we would encourage you to submit a revised manuscript to PLOS ONE.

Best regards,

Dr. Donovan McGrowder

Reviewers' comments:

Reviewer's Responses to Questions

**Comments to the Author**

1. Is the manuscript technically sound, and do the data support the conclusions?

Reviewer #1: Yes

Reviewer #2: Yes

2. Has the statistical analysis been performed appropriately and rigorously? 

Reviewer #1: Yes

Reviewer #2: I Don't Know

3. Have the authors made all data underlying the findings in their manuscript fully available?

Reviewer #1: Yes

Reviewer #2: Yes

4. Is the manuscript presented in an intelligible fashion and written in standard English?

Reviewer #1: Yes

Reviewer #2: Yes

5. Review Comments to the Author

Reviewer #1: Dear Authors,

I would like to highlight the need for the articles like this which target the specificity of the population. The article is well written, the hypothesis is formulated clearly, and I identified only minor flaws that should be addressed.

Please, change the chemical parameters into biochemical parameters/values/test/reference, etc. in the whole article (it is widely used).

Abstract:

1. the glucose value does not have a unit - mg/dL

2. the analyzer Dimi CS-T240 does not need "chemistry" in its "name", please delete (the same in the material and methods part)

Sample size and sampling techniques:

1. please describe more concrete how the participants were selected - for example at the beginning of the study involved 1000 participants, after exclusion only 344 were suitable for the study.

2. Please describe/characterize better participants (not only age, sex), add e.g. BMI

3. do you think that 60 min clothing of blood samples at RT can affect the reached data?

4. in the article is mentioned that sampling was provided between 8 - 11 am. This is too big a time gate... Do you know the time of starvation of the participants who came for blood sampling at 11 am?

5. do you have any information about how long/far the probands have to travel (by car, by foot - run) to reach the sampling date?

Results:

1. A total of 325 (164 males and 164 females) - in the abstract is 344 participants... how?

Discussion:

I would appreciate the explanation of the obtained results more than a comparison.

Reviewer #2: A well written article, on an important subject.

Errors:

Table 1: U/L: units per litter -> units per Liter (also in Table 2 and Table 3)

AST - Females, 90% CI (lower limit) "5.0, 16.0" should propably be "15.0, 16.0"

It would be helpful, if the Discussion included the important distinction between a 95% reference interval determined in a specific, local group and relevant diagnostic thresholds. When all participants are apparently healthy, no conclusions can be drawn regarding diagnostic significant values.

It is important to determine the local "normal values", but also to allow for diversity, when setting thresholds for diagnostic significance (hopefully the clinicians also recognise this).

---

## [Author Response · Author response to Decision Letter 0]

12 Oct 2022

Response to Journal Requirements

Comment # 1: Please ensure that your manuscript meets PLOS ONE's style requirements, including those for file naming. The PLOS ONE style templates can be found at https://journals.plos.org/plosone/s/file?id=wjVg/PLOSOne_formatting_sample_main_body.pdf and https://journals.plos.org/plosone/s/file?id=ba62/PLOSOne_formatting_sample_title_authors_affiliations.pdf

Response #1: As suggested, our manuscript follows PLOS ONE formatting to meet PLOS ONE's style requirements

Comment # 2: We suggest you thoroughly copyedit your manuscript for language usage, spelling, and grammar. If you do not know anyone who can help you do this, you may wish to consider employing a professional scientific editing service. 

Response #2: We do not know anyone who can help us do this, and we did not wish to consider employing a professional scientific editing service

Comment # 3: We note that you have stated that you will provide repository information for your data at acceptance. Should your manuscript be accepted for publication, we will hold it until you provide the relevant accession numbers or DOIs necessary to access your data. If you wish to make changes to your Data Availability statement, please describe these changes in your cover letter and we will update your Data Availability statement to reflect the information you provide.

Response #3: As stated in the Data Availability statement, we will provide repository information for our data at acceptance as Supporting Information files.

Response to Reviewer Comments 

Reviewer #1 

I would like to highlight the need for the articles like this which target the specificity of the population. The article is well written, the hypothesis is formulated clearly, and I identified only minor flaws that should be addressed.

Comment # 1: Please, change the chemical parameters into biochemical parameters/values/test/reference, etc. in the whole article (it is widely used).

Response #1: The term chemical parameters is not used in the manuscript; rather clinical chemistry parameters 

Abstract:

Comment # 1: the glucose value does not have a unit - mg/dL

Response #1: As suggested, it is stated as 65-125 mg/dL Line (37)

Comment # 2: the analyzer Dimi CS-T240 does not need "chemistry" in its "name", please delete (the same in the material and methods part)

Response #2: As suggested, it is deleted and stated as Dirui CS-T240 auto-analyzer Line (32) and Dirui CS T240 auto-analyzer Line (133) in the material and methods part

Sample size and sampling techniques:

Comment # 1: please describe more concrete how the participants were selected - for example at the beginning of the study involved 1000 participants, after exclusion only 344 were suitable for the study.

Response #1: As stated in the sample size and sampling techniques section, the study population was selected using the priori convenient sampling technique Line (99-101); i.e., an a priori sampling method was used where individuals had to meet well-defined exclusion/inclusion criteria before being selected as a referent individual. As suggested, it is stated as Considering altitude and residence difference; Dessie town, and Tita, Gerado and Borumeda rural communities were taken purposively from the study area. The determined sample size was distributed proportional to their population size of each selected community. From the selected communities, sub-communities were determined conveniently on the bases of easy to reach and suitability for blood sample transportation to the hospital laboratory, where biochemical analysis and serological testing was conducted. Finally, individuals who met the study eligibility criteria were included in the study from each community until the required number was achieved. Line (101-108) in the material and methods section and Of 344 participants consented for the study through a priori convenient sampling technique, 16 were excluded based on serological tests. A total 328 (164 males and 164 females) apparently healthy adults were included in the final analysis for establishing Line (167-169) in the result section.

Comment # 2: Please describe/characterize better participants (not only age, sex), add e.g. BMI

Response #2: As already stated in the manuscript, we conducted this study to establish clinical chemistry parameters reference intervals for apparently healthy adults and to investigate the sex-related difference in the reference intervals; we therefore described participants on the bases of age and sex. As suggested, it is stated as Majority of (59.4%) the participants were from urban communities and the mean body mass index was 23.2 ± 3.1 Kg/m2. Line (172-173) in the result section.

Comment # 3: do you think that 60 min clothing of blood samples at RT can affect the reached data?

Response #3: No. Serum provides the liquid portion of the blood without cells and clotting factors and, therefore, should contain proteins and other molecules that represent the whole body system. The cells and clotting factors must be removed from the blood sample by allowing adequate time for a clot to form. Most manufacturers of collections systems for serum samples recommend 3060 min at room temperature for a clot to form. (BD Technical Services News Bulletin). Centrifuging samples that is not completely clotted may cause the formation of fibrin in the serum, which may render the sample unusable for testing. Samples that sit longer than 60 min are likely to experience lysis of cells in the clot, releasing cellular components not usually found in serum samples. (Timms et al, 2007).

Comment # 4: in the article is mentioned that sampling was provided between 8 - 11 am. This is too big a time gate... Do you know the time of starvation of the participants who came for blood sampling at 11 am?

Response #4: As stated in the manuscript, blood samples were collected in the morning after an overnight fast from 8 to 11 am. Line (128-129); i.e., the time of fasting of the participants who came for blood sampling was at least 8 hours of fasting. 

Comment # 5: do you have any information about how long/far the probands have to travel (by car, by foot - run) to reach the sampling date?

Response #5: As stated in response #1 above, reference individuals in the community were included conveniently on the bases of easy to reach and suitability for blood sample transportation to the hospital laboratory.

Results:

Comment # 1: A total of 325 (164 males and 164 females) - in the abstract is 344 participants... how?

Response #1: In the result section it is stated as A total of 328 (164 males and 164 females) apparently healthy adults were included in the final analysis for establishing Line (168-169), not A total of 325 (164 males and 164 females) and also, in the abstract section is 328 apparently healthy adults Line (30), not 344 participants.... Regarding the reason for 328 reference individuals out of 344 participants, as stated in response #1 above, it is stated as Of 344 participants consented for the study through a priori convenient sampling technique, 16 were excluded based on serological tests. A total of 328 (164 males and 164 females) apparently Line (167-168)

Discussion:

Comment #: I would appreciate the explanation of the obtained results more than a comparison.

Response #: The explanation of the obtained results compared to other findings were given for each clinical chemistry parameter, including the explanation of the sex-related differences. The explanation of the obtained differences in the reference intervals of ALT and AST in this study compared to other study findings is given in Line (274-277); for ALP reference intervals Line (284-287); for total protein, total bilirubin and glucose reference intervals Line (305-307); for total cholesterol and triglycerides reference intervals Line (311-314); for renal function parameters (creatinine and urea) reference intervals Line (325-327).

Reviewer #2: 

A well written article, on an important subject.

Errors:

Comment # 1: Table 1: U/L: units per litter -> units per Liter (also in Table 2 and Table 3)

Response #1: As suggested, it is stated as Table 1: U/L: units per liter Line (205); Table 2: U/L: units per liter Line (239); and Table 3: U/L: units per liter Line (248).

Comment # 2: AST - Females, 90% CI (lower limit) "5.0, 16.0" should propably be "15.0, 16.0"

Response #2: It is typical error and it is stated as 15.0,16.0 Table 1: Line (202).

Comment # 3: It would be helpful, if the Discussion included the important distinction between a 95% reference interval determined in a specific, local group and relevant diagnostic thresholds. When all participants are apparently healthy, no conclusions can be drawn regarding diagnostic significant values.

Response #3: As already stated in the abstract and introduction sections of the manuscript, our aim was to establish reference intervals for commonly used clinical chemistry parameters not diagnostic thresholds (also known as clinical decision limits, action limits, cut-off points). Reference intervals are based on measurements in healthy individuals, diagnostic thresholds (decision limits) on measurements in patients. There is a clear distinction between healthy reference values measured in healthy populations or individuals and patient reference values measured in patients having various diseases. Yes, when all participants are apparently healthy, no conclusions can be drawn regarding diagnostic significant values (clinical decision limits). 

Comment # 4: It is important to determine the local "normal values", but also to allow for diversity, when setting thresholds for diagnostic significance (hopefully the clinicians also recognise this).

Response #4: As stated in response #3, our aim was to establish reference intervals not setting thresholds (decision limits) for diagnostic significance.

---

## [Editor Report · Decision Letter 1]

14 Oct 2022

Reference intervals for common clinical chemistry parameters in healthy adults of Northeast Ethiopia

PONE-D-22-12892R1

Dear Dr. Fiseha, 

We’re pleased to inform you that your manuscript has been judged scientifically suitable for publication and will be formally accepted for publication once it meets all outstanding technical requirements.

Kind regards,

Donovan Anthony McGrowder, PhD., MA., MSc

Academic Editor

PLOS ONE

Additional Editor Comments:

Dear Dr. Fiseha,

The manuscript was revised in accordance with the reviewers’ comments and is provisionally accepted pending final checks for formatting and technical requirements.

Best regards,

Dr. Donovan McGrowder (Academic Editor)

---

## [Editor Report · Acceptance letter]

21 Oct 2022

PONE-D-22-12892R1 

Reference intervals for common clinical chemistry parameters in healthy adults of Northeast Ethiopia 

Dear Dr. Fiseha:

I'm pleased to inform you that your manuscript has been deemed suitable for publication in PLOS ONE. Congratulations! Your manuscript is now with our production department. 

Kind regards, 

on behalf of

Dr. Donovan Anthony McGrowder 

Academic Editor

PLOS ONE